# Imbalanced Gradients: A New Cause of Overestimated Adversarial Robustness

## Abstract

Evaluating the robustness of a defense model is a challenging task in adversarial robustness research. Obfuscated gradients, a type of gradient masking, have previously been found to exist in many defense methods and cause a false signal of robustness. In this paper, we identify a more subtle situation called *Imbalanced Gradients* that can also cause overestimated adversarial robustness. The phenomenon of imbalanced gradients occurs when the gradient of one term of the margin loss dominates and pushes the attack towards to a suboptimal direction. To exploit imbalanced gradients, we formulate a *Margin Decomposition (MD)* attack that decomposes a margin loss into individual terms and then explores the attackability of these terms separately via a two-stage process. We examine 12 state-of-the-art defense models, and find that models exploiting label smoothing easily cause imbalanced gradients, and on which our MD attacks can decrease their PGD robustness (evaluated by PGD attack) by over 23%. For 6 out of the 12 defenses, our attack can reduce their PGD robustness by at least 9%. The results suggest that imbalanced gradients need to be carefully addressed for more reliable adversarial robustness.

## 1 Introduction

Deep neural networks (DNNs) are vulnerable to adversarial examples, which are input instances crafted by adding small adversarial perturbations to natural examples. Adversarial examples can fool DNNs into making false predictions with high confidence, and transfer across different models (Szegedy et al., 2014; Goodfellow et al., 2015). A number of defenses have been proposed to overcome this vulnerability. However, a concerning fact is that many defenses have been quickly shown to have undergone incorrect or incomplete evaluation (Carlini and Wagner, 2017; Athalye et al., 2018; Engstrom et al., 2018; Uesato et al., 2018; Mosbach et al., 2018; He et al., 2018). One common pitfall in adversarial robustness evaluation is the phenomenon of gradient masking (Papernot et al., 2017; Tramèr et al., 2018) or obfuscated gradients (Athalye et al., 2018), leading to weak or unsuccessful attacks and false signals of robustness. To demonstrate "real" robustness, newly proposed defenses claim robustness based on results of white-box attacks such as PGD (Madry et al., 2018), and at the same time, demonstrate that they are not a result of obfuscated gradients. In this work, we show that the robustness may still be overestimated even when there are no obfuscated gradients. Specifically, we identify a new situation called *Imbalanced Gradients* that exists in several state-of-the-art defense models and can cause highly overestimated robustness.

Imbalanced gradients is a new type of gradient masking effect where the gradient of one loss term dominates that of other terms. This causes the attack to move toward a suboptimal direction. Different from obfuscated gradients, imbalanced gradients are more subtle and are not detectable by the detection methods used for obfuscated gradients. To exploit imbalanced gradients, we propose a novel attack named *Margin Decomposition (MD)* attack that decomposes the margin loss into two separate terms, and then exploits the attackability of these terms via a two-stage attacking process. We derive MD variants of traditional attacks like PGD and MultiTargeted (MT) (Gowal et al., 2019), and deploy these MD attacks to re-examine the robustness of 12 adversarial training-based defense models. We find that 6 of them are susceptible to imbalanced gradients, and their robustness originally evaluated by the PGD attack drops significantly against our MD attacks. Our key contributions are:

- We identify a new type of subtle effect called *imbalanced gradients*, which can cause highly overestimated adversarial robustness and cannot be detected by detection methods

for obfuscated gradients. Especially, We highlight that label smoothing is one of the major causes of imbalanced gradients.

- We propose *Margin Decomposition (MD)* attacks to exploit imbalanced gradients. MD leverages the attackability of the individual terms in the margin loss in a two-stage attacking process. We also introduce two variants of MD for existing attacks PGD and MT.

- We conduct extensive evaluations on 12 state-of-the-art defense models and find that 6 of them suffer from imbalanced gradients and their PGD robustness drops by more than 9% against our MD attacks. Our MD attacks exceed state-of-the-art attacks when imbalanced gradients occur.

## 2 BACKGROUND

We denote a clean sample by $\mathbf{x}$, its class by $y \in \{1, \cdots, C\}$ with $C$ the number of classes, and a DNN classifier by $f$. The probability of $\mathbf{x}$ being in the $i$-th class is computed as $\mathbf{p}_i(\mathbf{x}) = e^{\mathbf{z}_i} / \sum_{j=1}^{C} e^{\mathbf{z}_j}$, where $\mathbf{z}_i$ is the logits for the $i$-th class. The goal of adversarial attack is to find an adversarial example $\mathbf{x}_{adv}$ that can fool the model into making a false prediction (*e.g.* $f(\mathbf{x}_{adv}) \neq y$), and is typically restricted to be within a small $\epsilon$-ball around the original example $\mathbf{x}$ (*e.g.* $\|\mathbf{x}_{adv} - \mathbf{x}\|_\infty \leq \epsilon$).

**Adversarial Attack.** Adversarial examples can be crafted by maximizing a classification loss $\ell$ by one or multiple steps of adversarial perturbations. For example the one-step Fast Gradient Sign Method (FGSM) (Goodfellow et al., 2015) and the iterative FGSM (I-FGSM) attack (Kurakin et al., 2017). Projected Gradient Descent (PGD) (Madry et al., 2018) attack is another iterative method that projects the perturbation back onto the $\epsilon$-ball centered at $\mathbf{x}$ when it goes beyond. Carlini and Wagner (CW) (Carlini and Wagner, 2017) attack generates adversarial examples via an optimization framework. Whilst there exist other attacks such as Frank-Wolfe attack (Chen et al., 2018a), distributionally adversarial attack (Zheng et al., 2019) and elastic-net attacks (Chen et al., 2018b), the most commonly used attacks for robustness evaluations are FGSM, PGD, and CW.

Several recent attacks have been proposed to produce more accurate robustness evaluations than PGD. This includes Fast Adaptive Boundary Attack (FAB) (Croce and Hein, 2019), MultiTargeted (MT) attack (Gowal et al., 2019), Output Diversified Initialization (ODI) attack (Tashiro et al., 2020), and AutoAttack (AA) (Croce and Hein, 2020). FAB finds the minimal perturbation necessary to change the class of a given input. MT (Gowal et al., 2019) is a PGD-based attack with multiple restarts and picks a new target class at each restart. ODI provides a more effective initialization strategy with diversified logits. AA attack is a parameter-free ensemble of four attacks: FAB, two Auto-PGD attacks, and the black-box Square Attack (Andriushchenko et al., 2019). AA has demonstrated to be one of the state-of-the-art attacks to date (Croce and Hein, 2020).

**Adversarial Loss.** Many attacks use Cross Entropy (CE) as the adversarial loss: $\ell_{ce}(\mathbf{x}, y) = -\log \mathbf{p}_y$. The other commonly used adversarial loss is the margin loss (Carlini and Wagner, 2017): $\ell_{margin}(\mathbf{x}, y) = \mathbf{z}_{max} - \mathbf{z}_y$, with $\mathbf{z}_{max} = \max_{i \neq y} \mathbf{z}_i$. Shown in (Gowal et al., 2019), CE can be written in a margin form (*e.g.* $\ell_{ce}(\mathbf{x}, y) = \log(\sum_{i=1}^{C} e^{\mathbf{z}_i}) - \mathbf{z}_y$), and in most cases, they are both effective. While FGSM and PGD attacks use the CE loss, CW and several recent attacks such as MT and ODI adopt the margin loss. AA has one PGD variant using the CE loss and the other PGD variant using the Difference of Logits Ratio (DLR) loss. DLR can be regarded as a "relative margin" loss. In this paper, we identify a new effect that causes overestimated adversarial robustness from the margin loss perspective and propose new attacks by decomposing the margin loss.

**Adversarial Defense.** In response to the threat of adversarial attacks, many defenses have been proposed such as defensive distillation (Papernot et al., 2016), feature/subspace analysis (Xu et al., 2017; Ma et al., 2018), denoising techniques (Guo et al., 2018; Liao et al., 2018; Samangouei et al., 2018), robust regularization (Gu and Rigazio, 2014; Tramèr et al., 2018; Ross and Doshi-Velez, 2018), model compression (Liu et al., 2018; Das et al., 2018; Rakin et al., 2018) and adversarial training (Goodfellow et al., 2015; Madry et al., 2018). Among them, adversarial training via robust min-max optimization has been found to be the most effective approach (Athalye et al., 2018). A number of new techniques have been proposed to further enhance the adversarial training (Wang et al., 2019; Zhang et al., 2019; Carmon et al., 2019; Alayrac et al., 2019; Wang and Zhang, 2019; Zhang and Wang, 2019; Zhang and Xu, 2020; Wang et al., 2020; Kim and Wang, 2020; Ding et al.,

2018; Chan et al., 2020). We will discuss and evaluate these adversarial training-based defenses with our proposed attacks in Section 4.

## 3 IMBALANCED GRADIENTS AND MARGIN DECOMPOSITION ATTACK

We first give a toy example of imbalanced gradients and show how regular attacks can fail in such a situation. We then empirically verify their existence in deep neural networks, particularly for some adversarially-trained models. Finally, we propose the Margin Decomposition attack to exploit the imbalanced gradients. Since CE and margin loss are the two commonly used loss functions for adversarial attack and CE can be written in a margin form (Gowal et al., 2019), here we focus on the margin loss to present the phenomenon of imbalanced gradients.

**Imbalanced Gradients.** The gradient of the margin loss (*e.g.* $\ell_{margin}(\mathbf{x}, y) = \mathbf{z}_{max} - \mathbf{z}_y$) is the combination of the gradients of its two individual terms (*e.g.* $\nabla_{\mathbf{x}}(\mathbf{z}_{max} - \mathbf{z}_y) = \nabla_{\mathbf{x}}\mathbf{z}_{max} + \nabla_{\mathbf{x}}(-\mathbf{z}_y)$). *Imbalanced Gradients* is the situation where the gradient of one loss term dominates that of other term(s), pushing the attack towards a suboptimal direction.

**Toy Example.** Consider a one-dimensional classification task and a binary classifier with two outputs $\mathbf{z}_1$ and $\mathbf{z}_2$ (like logits of a DNN), Figure 1 illustrates the distributions of $\mathbf{z}_1$, $\mathbf{z}_2$ and $\mathbf{z}_2 - \mathbf{z}_1$ around $x = 0$. The classifier predicts class 1 when $\mathbf{z}_1 \geq \mathbf{z}_2$, otherwise class 2. We consider an input at $x = 0$ with correct prediction $y = 1$, and a maximum perturbation constraint $\epsilon = 2$ (*e.g.* perturbation $\delta \in [-2, +2]$). The attack is successful if and only if $\mathbf{z}_2 > \mathbf{z}_1$. In this example, imbalanced gradients occurs at $x = 0$, where the gradients of the two terms $\nabla_x\mathbf{z}_2$ and $\nabla_x(-\mathbf{z}_1)$ have opposite directions, and the attack is dominated by the $\mathbf{z}_1$ term as $\nabla_x(-\mathbf{z}_1)$ is significantly larger than $\nabla_x\mathbf{z}_2$. Thus, attacking $x$ with the margin loss will converge to +2, where the sample is still correctly classified. However, for a successful attack, $x$ should be perturbed towards -2. In this particular scenario, the gradient $\nabla_x\mathbf{z}_2 < 0$ alone can provide the most effective attack direction. Note that this toy example was motivated by the loss landscape of DNNs when imbalanced gradient occurs.

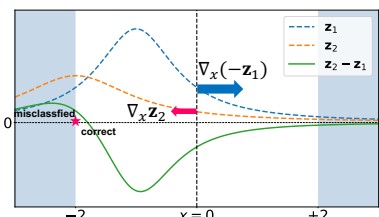

Figure 1: A toy illustration of *imbalanced gradients* at $x = 0$: the gradient of margin loss ($\mathbf{z}_2 - \mathbf{z}_1$) is dominated by its $-\mathbf{z}_1$ term, pointing to a suboptimal attack direction towards +2, where $x$ is still correctly classified.

### 3.1 IMBALANCED GRADIENTS IN DNNS

The situation can be extremely complex for DNNs with high-dimensional inputs, as imbalanced gradients can occur at each input dimension. It thus requires a metric to quantitatively measure the degree of gradient imbalance. Here, we propose such a metric named *Gradient Imbalance Ratio* (GIR) to measure the imbalance ratio for a single input $\mathbf{x}$, which can then be averaged over multiple inputs to produce the imbalance ratio for the entire model.

**Definition of GIR.** To measure the imbalance ratio, we focus on the input dimensions that are dominated by one loss term. An input dimension $x_i$ is dominated by a loss term (*e.g.* $\mathbf{z}_{max}$) means that 1) the gradients of loss terms at $x_i$ have different directions ($\nabla_{x_i}\mathbf{z}_{max} \cdot \nabla_{x_i}(-\mathbf{z}_y) < 0$), and 2) the gradient of the dominant term is larger (*e.g.* $|\nabla_{x_i}\mathbf{z}_{max}| > |\nabla_{x_i}(-\mathbf{z}_y)|$). According to the dominant term, we can split these dimensions into two subsets $\mathbf{x}_{s_1}$ and $\mathbf{x}_{s_2}$ where $\mathbf{x}_{s_1}$ are dominated by the $\mathbf{z}_{max}$ term, while $\mathbf{x}_{s_2}$ are dominated by the $-\mathbf{z}_y$ term. The overall dominance effect of each loss term can be formulated as $r_1 = \left\|\nabla_{\mathbf{x}_{s_1}}(\mathbf{z}_{max} - \mathbf{z}_y)\right\|_1$ and $r_2 = \left\|\nabla_{\mathbf{x}_{s_2}}(\mathbf{z}_{max} - \mathbf{z}_y)\right\|_1$. Here, we use the $L_1$-norms instead of $L_0$-norms (*i.e.* the number of dominated dimensions) to also take into consideration the gradient magnitude. To keep the ratio larger than 1, GIR is computed as:

$$GIR = \max\{\frac{r_1}{r_2}, \frac{r_2}{r_1}\} \tag{1}$$

GIR is defined by the ratio between the $L_1$-norms of the gradients of two groups:

Note that the GIR metric is not a general measure of imbalance. Rather, it is designed only for assessing gradient imbalance for *adversarial robustness evaluation*. GIR focuses specifically on the

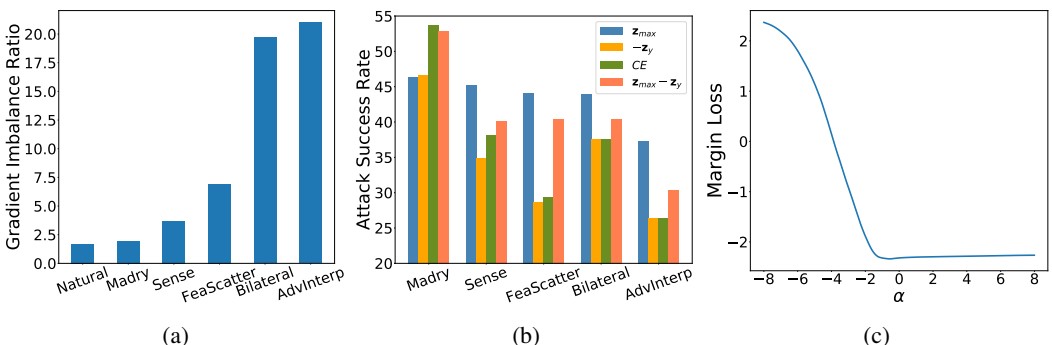

Figure 2: (a): Gradient imbalance ratio of 5 models. (b): Attack success rate of PGD-20 with different losses. (c): The margin loss of the AdvInterp defense model on points $\mathbf{x}^* = \mathbf{x} + \alpha \cdot \text{sign}(\nabla_{\mathbf{x}}(-\mathbf{z}_y))$, where $\mathbf{x}$ is a natural sample and $\text{sign}(\nabla_{\mathbf{x}}(-\mathbf{z}_y))$ is the signed gradient of loss term $-\mathbf{z}_y$. All these experiments are conducted on test images of CIFAR-10.

imbalanced input dimensions, and uses the $L_1$ norm to also take into account the influence of these dimensions to the final output. The ratio reflects how far away the imbalance towards one direction than the other.

**GIR of both Naturally- and Adversarial-trained DNNs.** With the GIR metric, we next investigate 6 DNN models including a naturally-trained (Natural) model and 5 adversarially-trained models using standard adversarial training (Madry et al., 2018) (SAT), sensible adversarial training (Kim and Wang, 2020) (Sense), feature scattering-based adversarial training (Zhang and Wang, 2019) (FeaScatter), bilateral adversarial training (Wang and Zhang, 2019) (Bilateral), and adversarial interpolation training (Zhang and Xu, 2020) (AdvInterp). We present these defense models here because they represent different levels of gradient imbalance (a complete analysis of more models can be found in Appendix F). Natural, SAT and Sense are WideResNet-34-10 models, while others are WideResNet-28-10 models. We train Natural and SAT following typical settings in (Madry et al., 2018) while others use their officially released models. We compute the GIR scores of the 6 models based on 1000 randomly selected test samples, and show them in Figure 2a. One major observation is that some defense models can have a much higher imbalance ratio than either naturally-trained or SAT model. This confirms that gradient imbalance does exist in DNNs, and some defenses tend to train the model to have highly imbalanced gradients. We will show, in Section 4, that this situation of imbalanced gradients may cause highly overestimated robustness when evaluated using a traditional PGD attack.

**Imbalanced Gradients Reduce Attack Effectiveness.** When there are imbalanced gradients, the attack can be pushed by the dominant term to produce weak attacks, and the non-dominant term alone can lead to more successful attacks. To illustrate this, in Figure 2b, we show the success rates of PGD attack on the above 5 defense models (Natural has zero robustness against PGD) with different losses: CE loss, margin loss, and the two individual margin terms. We consider 20-step PGD (PGD-20) attacks with step size $\epsilon/4$ and $\epsilon = 8/255$ on all CIFAR-10 test images. Intuitively, the two margin terms could lead to less effective attacks, as they only provide partial information about the margin loss. This is indeed the case for the low gradient imbalance model SAT. However, for highly imbalanced models Sense, FeaScatter, Bilateral and AdvInterp, attacking the $\mathbf{z}_{max}$ term produces even more powerful attacks than attacking the margin loss. This indicates that the gradient of the margin loss is shifted by the dominant term (e.g. $-\mathbf{z}_y$ in this case) towards a less optimal direction, which inevitably causes less powerful attacks. Compared between attacking CE loss and attacking $-\mathbf{z}_y$, they achieve a very close performance on imbalanced models. This shows a stronger dominant effect of $-\mathbf{z}_y$ in CE loss ($\ell_{ce}(\mathbf{x}, y) = \log(\sum_{i=1}^{C} e^{\mathbf{z}_i}) - \mathbf{z}_y$). It is worth mentioning that, while both GIR and this individual term-based test can be used to check whether there are significantly imbalanced gradients in a defense model, GIR alone cannot fully reflect the attack success rate. Figure 2c shows an example of how the $-\mathbf{z}_y$ term leads the attack to a suboptimal direction: the margin loss is flat at the $\nabla_{\mathbf{x}}(-\mathbf{z}_y)$ direction, yet increases drastically at an opposite direction. In this example, the attack can actually succeed if it increases (rather than decreases) $\mathbf{z}_y$.

**Gradients can be Balanced by Attacking Individual Loss Terms.** Here, we show that, interestingly, imbalanced gradients can be balanced by attacking the non-dominant term. Consider the AdvInterp model tested above as an example, the dominant term is $-\mathbf{z}_y$. Figure 3 illustrates the

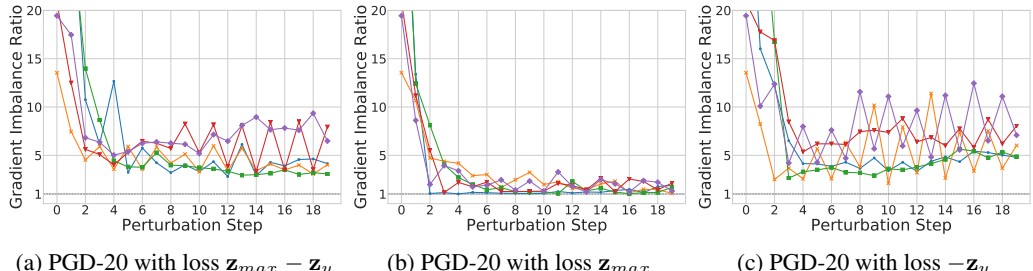

(a) PGD-20 with loss $\mathbf{z}_{max} - \mathbf{z}_y$     (b) PGD-20 with loss $\mathbf{z}_{max}$     (c) PGD-20 with loss $-\mathbf{z}_y$

Figure 3: Changes in gradient imbalance ratio when apply PGD-20 ($\epsilon = \frac{8}{255}$) attack with the margin loss (a), only the $\mathbf{z}_{max}$ term (b), or only the $-\mathbf{z}_y$ term (c), on the AdvInterp model for 5 CIFAR-10 test images. The imbalance ratio is effectively reduced by attacking a single $\mathbf{z}_{max}$ term.

GIR values of 5 randomly selected CIFAR-10 test images by attacking them using PGD-20 with different margin terms or the full margin loss. As can be observed that, for all three losses, the GIRs are effectively reduced after the first few steps. However, only the non-dominant term $\mathbf{z}_{max}$ manages to stably reduce the imbalance ratio to around 2. This indicates optimizing the individual terms separately can help avoid the situation of imbalanced gradients and the attack can indeed benefit from more balanced gradients (see the higher success rate of $\mathbf{z}_{max}$ in Figure 2b).

## 3.2 MARGIN DECOMPOSITION ATTACK

The above observations motivate us to exploit the individual terms in the margin loss so that the imbalanced gradients situation can be circumvented. Specifically, we propose Margin Decomposition (MD) attack that decomposes the attacking process with a margin loss into two stages: 1) alternately attacking the two individual terms (*e.g.* $\mathbf{z}_{max}$ or $-\mathbf{z}_y$) at different restarts; then 2) attacking the full margin loss. Formally, our MD attack and its loss functions in each stage is defined as follows:

$$\mathbf{x}_{k+1} = \Pi_\epsilon(\mathbf{x}_k + \alpha \cdot \text{sign}(\nabla_\mathbf{x} \ell_k^r(\mathbf{x}_k, y))), \tag{2}$$

$$\ell_k^r(\mathbf{x}_k, y) = \begin{cases} \mathbf{z}_{max} & \text{if } k < \frac{K}{2} \text{ and } r \bmod 2 = 0 \\ -\mathbf{z}_y & \text{if } k < \frac{K}{2} \text{ and } r \bmod 2 = 1 \\ \mathbf{z}_{max} - \mathbf{z}_y & \text{if } k \geq \frac{K}{2}, \end{cases}$$

where, $\Pi$ is the projection operation that projects the perturbed sample back within $\epsilon$-ball, $k \in \{1, \ldots, K\}$ is the perturbation step, $r \in \{1, \ldots, n\}$ is the $r$-th restart, $\bmod$ is the modulo operation for alternating optimization, and $\ell_k^r$ defines the loss function used at the $k$-th step and $r$-th restart. The loss function switches from the individual terms back to the full margin loss at step $\frac{K}{2}$. The first stage exploits individual loss terms to rebalance the imbalanced gradients, while the second stage ensures that the final objective (*e.g.* maximizing the classification error) is achieved. Note that, not all defense models have the imbalanced gradients problem. A model is susceptible to imbalanced gradients if there is a substantial difference between robustness evaluated by PGD attack and that by our MD attack. In addition, to help escape the flat loss landscape observed in Figure 2c, we initialize the perturbation in the first stage by perturbing one step with size $2 \cdot \epsilon$ along the opposite direction of the other loss terms that are left unexplored.

We also propose a Margin Decomposition Multi-Targeted (MDMT) attack, a multi-targeted version of our MD attack. The loss terms used by MDMT at different attacking stages are defined as follows:

$$\ell_k^r(\mathbf{x}_k, y) = \begin{cases} \mathbf{z}_t & \text{if } k < \frac{K}{2} \text{ and } r \bmod 2 = 0 \\ -\mathbf{z}_y & \text{if } k < \frac{K}{2} \text{ and } r \bmod 2 = 1 \\ \mathbf{z}_t - \mathbf{z}_y & \text{if } k \geq \frac{K}{2}, \end{cases}$$

where, $\mathbf{z}_t$ is the logits of the target class $t \neq y$. Like the MT attack, MDMT will attack each possible target class one at a time, then select the strongest adversarial example at the end. That is, the target class $t \neq y$ will be switched to a different target class at each restart. The complete algorithms of MD and MDMT can be found in Appendix A, and an ablation study can be found in Appendix G.

**Initialization Perspective Interpretation of MD Attacks**. Previous works have shown that random or logits diversified initialization are crucial for generating strong adversarial attacks (Madry et al., 2018; Tashiro et al., 2020). Compared to random or logits diversified initialization, our MD attacks

can be interpreted as a type of *adversarial initialization*, *i.e.*, initialize at the adversarial sub-directions defined by the two terms of the margin loss. Moreover, rather than a single step of initialization, our MD attacks iteratively explore the optimal starting point during the entire first attacking stage.

## 4    EXPERIMENTS

We apply our MD attacks to evaluate the robustness of 12 state-of-the-art defense models. We focus on adversarial training models, which are arguably the strongest defense approaches to date (Athalye et al., 2018; Croce and Hein, 2020). All the models are WideResNet variants (Zagoruyko and Komodakis, 2016) and are trained against perturbation $\epsilon = 8/255$ on CIFAR-10. For each defense model, we either download their shared models or retrain the models using the official implementations, unless explicitly stated. Further details about the models can be found in Appendix E. We apply current state-of-the-art attacks and our MD attacks to evaluate the robustness of these models in a white-box setting.

**Baseline Attacks and Settings.** Following the current literature, we consider 6 existing attacks: 1) FGSM, 2) PGD, 3) $L_\infty$ version of CW attack (Madry et al., 2018; Wang et al., 2019), 4) MultiTargeted (MT) attack and two concurrently proposed attackss 5) AutoAttack (AA), and 6) Output Diversified Initialization (ODI). The evaluation is done under the same maximum perturbation $\epsilon = 8/255$ for training. For AA and ODI, we use the official implementation and parameter setting. For regular iterative attacks, we set the step size to $\alpha = \epsilon/4$ and the total perturbation steps to $K = 40$. For our MD and MDMT, we use a large step size $\alpha = 2 \cdot \epsilon$ in the first stage for a better exploration and $\alpha = \epsilon/4$ in the second stage to ensure a stable optimization for the final objective. For regular attacks PGD, CW and our MD, we use 2 random restarts, while for more powerful attacks ODI, MT and MDMT, we use 20 restarts (MT attacks require more restarts to explore multiple target classes). A parameter analysis of our MD attack can be found in Appendix H. Adversarial robustness is measured by the model accuracy on adversarial examples crafted by these attacks on CIFAR-10 test images.

### 4.1    EVALUATION RESULTS

Table 1 reports the full evaluation result, where RST, UAT and TRADES are the top 3 best defenses. The SAT defense demonstrates $\sim 45\%$ robustness consistently against either PGD or stronger attacks such as MT, AA, ODI and our MD attacks. This indicates that SAT does not have imbalanced gradients and indeed brings consistent robustness, which is in line with other studies about SAT (Athalye et al., 2018; Croce and Hein, 2020; Uesato et al., 2018). While the rest 11 defense models are all developed based on SAT, they exhibit quite different robustness. Only 4 defenses including RST, UAT, TARDES and MART are indeed improved over SAT, while the other 7 defense models are actually not as robust as SAT, according to our MD or MDMT attacks. For the 4 improved defenses, their PGD robustness (*e.g.* robustness evaluated by PGD attack) can still be reduced by stronger attacks MT, AA, ODI or our MD attacks. Considering that their robustness drops against our MD attacks are within $5\%$, their drops may be caused by sufficient explorations such as more random restarts or better initialization rather than imbalanced gradients. Indeed, MT, AA, and ODI with more random restarts, multiple target classes, and better initialization can also reduce their robustness to the same level as our MD attacks.

Out of the 7 unimproved defenses, our MDMT attack can reduce the PGD robustness of 6 models (*e.g.* MMA, Bilateral, Adv-Interp, FeaScatter, Sense, and JARN-AT11) by at least 9%. On all 7 unimproved defenses, our MD attacks are always the most effective attacks compared to either classic attacks FGSM, PGD, CW, or more recent attacks MT, AA and ODI. Note that, for 4 (*e.g.* MMA, Bilateral, Adv-Interp, and Sense) out of the 7 unimproved defenses, even state-of-the-art attacks MT or AA evaluate them to be more robust than SAT, which is not necessarily the case according to our MD attacks. Particularly, against the MT attack, the robustness of SAT is 45.34%, while the robustness of Bilateral, Adv-Interp and Sense are 55.07%, 61.22% and 46.22%, respectively. For the MMA defense, AA attack evaluates its robustness to be 45.69%, which is slightly higher than SAT's 45.26%. However, under our MD attacks, all 4 models show much lower robustness than SAT (3%-10% lower). Next, we will investigate the imbalanced gradients problem in the unimproved defenses.

The PGD results with grid searched step size are reported in Appendix I, where it shows larger step size can help PGD when there are imbalanced gradients, yet is still far less effective than our MD

Table 1: Robustness (%) of 12 defense models evaluated by different attacks. The attacks are divided into 2 groups: 1) traditional attacks for robustness evaluation and our MD (column 3-6); and 2) more recent attacks and our MDMT (column 7-10). The defenses are also divided into 2 groups: 1) SAT or improved defenses (top rows); and 2) those that are not improved over SAT (bottom rows). Results in $(\cdot)$ in the MDMT column show the robustness decrease compared to the PGD attack.

| Defense | Clean | FGSM | PGD | CW | MD | MT | AA | ODI | MDMT |
|---|---|---|---|---|---|---|---|---|---|
| RST | 89.69 | 69.60 | 62.09 | 60.87 | **60.17** | 59.80 | 59.66 | 59.93 | 59.86 (-2.23) |
| UAT | 86.46 | 68.31 | 61.08 | 62.11 | **59.36** | 56.72 | 56.94 | 57.98 | 56.65 (-4.43) |
| TRADES | 84.92 | 60.87 | 55.00 | 53.69 | **53.10** | 52.67 | 53.18 | 52.68 | 52.78 (-2.22) |
| MART | 83.09 | 61.43 | 56.10 | 53.02 | **51.84** | 51.12 | **51.05** | 51.15 | 51.07 (-5.03) |
| SAT | 86.83 | 56.88 | 45.94 | 45.73 | **45.64** | 45.34 | **45.17** | 45.26 | 45.25 (-0.69) |
| Dynamic | 85.35 | 55.19 | 46.36 | 45.53 | **43.93** | 42.75 | 42.88 | 43.03 | **42.69** (-3.67) |
| MMA | 84.62 | 61.85 | 51.09 | 52.05 | **45.63** | 42.62 | 45.69 | 43.00 | **41.92** (-9.17) |
| Bilateral | 90.73 | 71.10 | 60.95 | 57.82 | **39.82** | 55.07 | 37.96 | 38.65 | **37.21** (-23.74) |
| Adv-Interp | 90.25 | 77.94 | 72.48 | 67.92 | **45.33** | 61.22 | 38.58 | 41.43 | **37.59** (-34.89) |
| FeaScatter | 89.98 | 77.40 | 68.64 | 57.10 | **43.12** | 43.10 | 38.79 | 39.61 | **36.86** (-31.78) |
| Sense | 91.51 | 72.71 | 59.86 | 57.67 | **40.64** | 46.22 | 36.10 | 38.15 | **35.25** (-24.61) |
| JARN-AT1 | 81.96 | 61.48 | 42.50 | 27.46 | **15.03** | 16.01 | 30.11 | 14.90 | **14.60** (-27.90) |

attacks. A comparison of our MDMT attack to the 4 individual attacks in the AA ensemble can be found in Appendix J, where it shows our MD attack is superior to any of the 4 individual attacks on 6 out of the 12 tested defense models, and our MD attack is the best across all defense models. More evaluation results on 3 defense models trained on CIFAR-100 can be found in Appendix K.

**Efficiency Analysis.** We compare the efficiency of the two best attacks identified in Table 1: our MDMT attack and AA attack. Ensemble attacks like AA are generally more powerful than standalone attacks, yet are also more time-consuming. To test this, here we also include the AA+[1] attack, which is an updated version of AA with an ensemble of 6 different attacks. We repeat the attack for 5 times on the entire CIFAR10 test set and report the average time cost in Table 2. The time cost is measured with respect to a single 2080TI GPU. As can be observed, our MDMT attack is at least 8 times more efficient than AA. Having two more attacks in the ensemble, AA+ is notably more time-consuming that the AA attack. The adaptation of our method to existing attacks only needs to replace the loss used in the first half of the iteration steps to the decomposed loss terms following Equation 2, thus does not increase the time complexity of the original attacks.

Table 2: Average time cost (in hours) of MTMD, AA and AA+ attacks on defense models Adv-Interp, FeaScatter and Sense over 5 repeats on the entire CIFAR-10 test set. The best results are in **bold**.

| Defense | MDMT | AA | AA+ |
|---|---|---|---|
| Adv-Interp | **2.01hrs** | 16.36hrs | 20.71hrs |
| FeaScatter | **1.97hrs** | 15.97hrs | 20.44hrs |
| Sense | **2.28hrs** | 18.35hrs | 23.44hrs |

## 4.2 DEFENSE TECHNIQUES THAT MAY CAUSE IMBALANCED GRADIENTS

Here, we focus on 6 unimproved (compared to SAT) defenses: MMA, Bilateral, Adv-Interp, FeaScatter, Sense, and JARN-AT1. Their PGD-evaluated robustness has been reduced for $> 9\%$ by our MDMT attack.

**Label Smoothing Causes Imbalanced Gradients.** The PGD robustness of Bilateral, FeaScatter, and Adv-Interp decrease the most (*e.g.* $23\% - 34\%$) against our MDMT attack. This indicates that these defenses may have caused imbalanced gradients, as also indicated by their high GIR values in Figure 2a. All three defenses use label smoothing as part of their training scheme to improve adversarial training, which we suspect is one common cause of imbalanced gradients. Given a sample $\mathbf{x}$ with label $y$, label smoothing encourages the model to learn an uniform logits or probability distribution over classes $j \neq y$. This tends to smooth out the input gradients of $\mathbf{x}$ with respect to these

---

[1]https://github.com/fra31/auto-attack

Table 3: Robustness (%) of WideResNet-34-10 models trained with/without label smoothing.

| **Defense** | FGSM | PGD | MD |
|---|---|---|---|
| SAT | 56.88 | 46.47 | **45.71** |
| + Label Smoothing | 59.10 | 51.15 | **44.54** |
| Natural | 26.41 | **0.00** | **0.00** |
| + Label Smoothing | 48.09 | 10.86 | **0.00** |

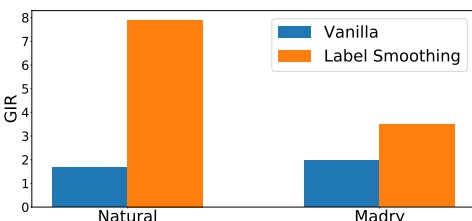

Figure 4: Gradient Imbalance Ratio (GIR) of models trained with/without label smoothing.

classes, resulting in smaller gradients. In order to confirm label smoothing indeed causes imbalanced gradients, we train a WideResNet-34-10 model using natural training ('Natural') and SAT with or without label smoothing (smoothing parameter 0.5). We report their robustness in Table 3, and show their gradient imbalance ratios (GIRs) in Figure 4. According to GIRs, adding label smoothing into the training process immediately increases the imbalance ratio, especially in natural training. The PGD robustness of the naturally-trained model also "increases" to 10.86%, which is still 0% under our MD attack. Using smoothed labels in SAT defense also "increases" PGD robustness by almost 5%, which in fact, decreases by 1%. These evidences confirm that label smoothing indeed causes imbalanced gradients, leading to overestimated robustness if evaluated by regular attacks like PGD. Interestingly, it appears that adversarial training can inhibit moderately the imbalanced gradients problem of label smoothing. This is because the adversarial examples used for adversarial training are specifically perturbed to the $j \neq y$ classes, thus helping avoid uniform logits over classes $j \neq y$ to some extent.

**Other Defense Techniques that may Cause Imbalanced Gradients.** The other 3 unimproved defenses MMA, Sense and JARN-AT1 adopt different defense techniques to improve robustness. MMA is a margin-based defense that maximizes the shortest successful perturbation for each data point. MMA only perturbs correctly classified clean examples, and the perturbation stops immediately at misclassification (into a $j \neq y$ class). In other words, MMA focuses on examples that are around the decision boundary (*e.g.* $\mathbf{z}_{max} = \mathbf{z}_y$) between class $y$ and all other classes $j \neq y$. During training, the decision boundary margin is maximized by pulling the boundary away from these examples. This process tries to maximize the distance to the closest decision boundary (*e.g.* towards the weakest class) and finally results in equal distances to all other classes. This tends to generate a uniform prediction over classes $j \neq y$, a similar effect of label smoothing, and causes imbalanced gradients.

Similar to MMA, Sense perturbs training examples until a certain loss threshold is satisfied. While in MMA the threshold is misclassification, in Sense, it is the loss value with respect to probability (*e.g.* $\mathbf{p}_y = 0.7$). This type of training procedures with a specific logits or probability distribution regularization has caused the imbalanced gradients problem for both MMA and Sense. Note that, Sense causes much severe imbalanced gradients than MMA. We conjecture it is because optimizing over a probability threshold is much easier than moving the decision boundary.

JARN-AT1 is also a regularization-based adversarial training method. Different from MMA or Sense, it regularizes the model's Jacobian (*e.g.* input gradients) to resemble natural training images. Such an explicit input gradients regularization tends to reduce the input gradients to a much smaller magnitude and only keep the salient part of input gradients. The input gradients associated with other $j \neq y$ classes will be minimized to cause an imbalance to that associated with class $y$. This has caused PGD to produce 27.90% more robustness than our MDMT attack. Note that, even the recent AA attack still produces 15.51% overestimated robustness compared to our MDMT.

**Correlation between GIR and Robustness.** According to the GIR scores shown in Figure 2a and Figure 6 (Appendix F), models exhibit high GIR scores (*e.g.* Adv-Interp, FeaScatter and Bilateral) are generally more prone to imbalanced gradients and are potentially more vulnerable to our MD attacks. However, GIR is not a measure for robustness nor should be used as an exact metric to determine whether one defense is more robust than the other. For example, MART demonstrates a higher GIR score than Sense or JARN-AT1, however, according to our MDMT attack, it is 15.82% and 36.47% more robust than Sense and JARN-AT1, respectively. This is because the GIR score of a model only measures the gradient situation of the model at its current state, which could decrease during the attack as shown in Figure 3 and 5 (Appendix C). Our MD attacks iteratively exploit and circumvent imbalanced gradients during the first attacking stage, thus can produce reliable robustness evaluation at the end.

### 4.3 An Attack View of Imbalanced Gradients

As shown in Table 1, recent attacks ODI and AA are more effective than traditional attacks PGD and CW against imbalanced gradients. Here, we provide some insights into why these techniques are effective against imbalanced gradients. We consider attacking AdvInterp as an example and show how the gradient imbalance ratio (GIR) changes in different attacking processes.

**Logits Diversified Initialization Helps Avoid Imbalanced Gradients.** ODI randomly initializes the perturbation by adding random weights to logits at its first 2 steps. The random weights change the gradients' size, thus can also mitigate imbalanced gradients, as shown in Figure 5a in Appendix C. However, initialization only helps the first 2 steps, and the imbalance ratio still jumps in the following steps. Our attack provides a more direct and efficient exploration of imbalance gradients, thus can maintain a low imbalance ratio even after the first few steps (see Figure 5c in Appendix C). As also shown in Table 1, our MDMT attack is consistently more effective than ODI.

**Exploration Beyond the $\epsilon$-ball Helps Avoid Imbalanced Gradients.** AA is an ensemble of four attacks: two Auto-PGD attacks and two existing attacks FAB and Square. By inspecting the individual attacks, we found that the most effective method is FAB. FAB first finds a successful attack using unbounded perturbation size (*e.g.* $> \epsilon$), then minimizes the perturbation to be within the $\epsilon$-ball. As shown in Figure 5b in Appendix C, the first few steps of exploration outside the $\epsilon$-ball can effectively avoid imbalanced gradients. This is also why our MD attacks use a large step size in the first stage. However, the imbalance ratio tends to increase when FAB attempts to minimize the perturbation (steps 10 - 16). We believe FAB can be further improved following our decomposition strategy.

**Imbalanced Gradients are not Easily Detected or Circumvented by Existing Methods.** We also show, in Appendix B, that defense models with imbalanced Gradients can still pass the five checking rules of obfuscated gradients, and that many times of restarts with random initialization or momentum method does not help escape imbalanced gradients in Appendix D. This makes imbalanced gradients more subtle and should be carefully checked to avoid overestimated robustness.

**Black-box Attacks can be Improved by Circumventing Imbalanced Gradients.** Here we show gradient estimation based black-box attacks can also benefit from our MD method when there are imbalanced gradients. We take SPSA as an example, and use the two-stage losses of our MD attack for SPSA. This version of SPSA is denoted as SPSA+MD. For both SPSA and SPSA+MD, we use the same batch size of 8192 with 100 iterations, and run on 1000 randomly selected CIFAR-10 test images. The attack success rates on Adv-Interp, FeaScatter and Sense models are reported in Table 4. Compared to SPSA, SPSA+MD can lower the robustness by at least 5.9%. This indicates that imbalanced gradient also has a negative impact on back-box attacks, and our method can be easily applied to produce more queries-efficient and successful black-box attacks.

Table 4: Attack success rate (ASR) of the SPSA attack with or without our MD on three CIFAR-10 defense models. The ASRs are tested on the entire CIFAR-10 test set. The best results are in **bold**.

| Attack | Adv-Interp | FeaScatter | Sense |
|--------|------------|------------|-------|
| SPSA | 24.80% | 28.29 | 37.90 |
| SPSA+MD | **40.30%** | **45.60%** | **48.80%** |

## 5 Conclusion

In this paper, we identify *Imbalanced Gradients*, a new situation where traditional attacks such as PGD can fail and produce overestimated adversarial robustness. We proposed a new metric to investigate the imbalanced gradients problem in current defense models. We also proposed a new attack called Margin Decomposition (MD) attack to leverage imbalanced gradients via a two-stage attacking process. By evaluating 12 state-of-the-art defense models, we find that 6 of them are susceptible to imbalanced gradients and their PGD robustness suffers a significant drop against our MD attacks. We identified a set of possible causes of imbalanced gradients, and effective countermeasures. Our results indicate that future defenses should avoid causing imbalanced gradients to obtain more reliable adversarial robustness.

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

# A    ALGORITHMS OF MT AND MDMT ATTACKS

Algorithm 1 and Algorithm 2 below describe the complete attacking procedure of our Margin Decomposition (MD) attack and its Multi-Targeted (MDMT) version.

---

**Algorithm 1** Margin Decomposition Attack

---

1: **Input:** clean sample $\mathbf{x}$, label $y$, model $f$.
2: **Output:** adversarial example $\mathbf{x}_{adv}$
3: **Parameters:** Perturbation bound $\epsilon$, step size $\alpha$, number of restarts $n$, number of steps $K$.
4: $\mathbf{x}_{adv} \leftarrow \mathbf{x}$
5: **for** $r \in \{1, ..., n\}$ **do**
6:     Initialize $\mathbf{x}_0$ by one step of perturbation along the opposite direction of gradients.
7:     **for** $k \in \{1, ..., K\}$ **do**
8:         Update $\mathbf{x}_k$ by Eq. (2)
9:         **if** $\ell(\mathbf{x}_{adv}) < \ell(\mathbf{x}_k)$ **then**
10:             $\mathbf{x}_{adv} \leftarrow \mathbf{x}_k$
11:         **end if**
12:     **end for**
13: **end for**
14: **return** $\mathbf{x}_{adv}$

---

---

**Algorithm 2** Margin Decomposition MultiTargeted attack

---

1: **Input:** clean sample $\mathbf{x}$, class label $y$, class set $\mathcal{T}$, model $f$.
2: **Output:** adversarial example $\mathbf{x}_{adv}$
3: **Parameters:** Perturbation bound $\epsilon$, PGD step size $\alpha$, number of restarts $n$, number of steps $K$.
4: $n_r \leftarrow \lfloor n/|\mathcal{T}| \rfloor$, $\mathbf{x}_{adv} \leftarrow \mathbf{x}$
5: **for** $r \in \{1, ..., n_r\}$ **do**
6:     **for** $t \in \mathcal{T}$ **do**
7:         Initialize $\mathbf{x}_0$ by one step of perturbation along the opposite direction of gradients.
8:         **for** $k \in \{1, ..., K\}$ **do**
9:             Update $\mathbf{x}_k$ by Eq. (**??**)
10:             **if** $\ell(\mathbf{x}_{adv}) < \ell(\mathbf{x}_k)$ **then**
11:                 $\mathbf{x}_{adv} \leftarrow \mathbf{x}_k$
12:             **end if**
13:         **end for**
14:     **end for**
15: **end for**
16: **return** $\mathbf{x}_{adv}$

---

# B    IMBALANCED GRADIENTS ARE DIFFERENT FROM OBFUSCATED GRADIENTS

Imbalanced gradients occur when one loss term dominating the attack towards a suboptimal gradient direction, which does not necessarily block gradient descent like obfuscated gradients. Therefore, it does not have the characteristics of obfuscated gradients, and can not be detected by the five checking rules for obfuscated gradients (Athalye et al., 2018). Here, we test all the five rules on the four defense models that exhibited significant imbalanced gradients: Adv-Interp, FeaScatter, Bilateral, and Sense. Note that all these models were trained and tested on CIFAR-10 dataset.

**One-step attacks perform better than iterative attacks**. When gradients are obfuscated, iterative attacks are more likely to get stuck in a local minima. To test this, we compare the success rate of one-step attack FGSM and iterative attack PGD in Table 5. We see that PGD outperforms FGSM consistently on all the four defense models, i.e., no obvious sign of obfuscated gradients.

**Unbounded attacks do not reach 100% success. Increasing distortion bound does not increase success.** Larger distortion bound gives the attacker more ability to attack. So, if gradients are not obfuscated, unbounded attack should reach 100% success rate. To test this, we run an "unbounded" PGD attack with $\epsilon = 1$. As shown in Table 5, all models are completely broken by this unbounded

attack. This again indicates that the overestimated robustness is caused by a different effect rather than obfuscated gradients.

**Black-box attacks are better than white-box attacks.** If a model is obfuscating gradients, it should fail to provide useful gradients in a small neighborhood. Therefore, using a substitute model should be able to evade the defense, as the substitute model was not trained to be robust to small perturbations. To test this, we run black-box transferred PGD attack on naturally trained substitute models. We find that all four defenses are robust to transferred attacks ("Transfer" in Table 5). We also attack the four defense models using gradient-free attack SPSA (Uesato et al., 2018). For SPSA, we use a batch size of 8192 with 100 iterations, and run on 1000 randomly selected CIFAR-10 test images. We confirm that SPSA cannot degrade their performance. None of these results indicate obfuscated gradients.

**Random sampling finds adversarial examples.** Brute force random search within some $\epsilon$-ball should not find adversarial examples when gradient-based attacks do not. Following (Athalye et al., 2018), we choose 1000 test images on which PGD fails. We then randomly sample $10^5$ points for each image from its $\epsilon = 8/255$-ball region, and check if any of them are adversarial. The results (*e.g.* "Random") shown in Table 5 confirms that random sampling cannot find an adversarial example when PGD does not.

All the above test results lead to one conclusion that the robustness of the four defenses is not a result of obfuscated gradients. This indicates that imbalanced gradients does not share the characteristics of obfuscated gradients, thus cannot be detected following the five test principles for obfuscated gradients. This makes adversarial robustness evaluation more difficult. Therefore, imbalanced gradients should be carefully addressed for more accurate robustness evaluation.

Table 5: Test of obfuscated gradients for four defense models that have significant imbalanced gradients following (Athalye et al., 2018): attack success rate (%) of different attacks. None of the above results indicates a clear sign of obfuscated gradients.

| Defense | FGSM | PGD | Unbounded | Transfer | SPSA | Random |
|---|---|---|---|---|---|---|
| Adv-Interp (Zhang and Xu, 2020) | 23.06 | 27.52 | 100.00 | 10.89 | 24.80 | 0.00 |
| FeaScatter (Zhang and Wang, 2019) | 22.60 | 31.36 | 100.00 | 11.11 | 28.20 | 0.00 |
| Bilateral (Wang and Zhang, 2019) | 28.90 | 39.05 | 100.00 | 9.23 | 36.00 | 0.00 |
| Sense (Kim and Wang, 2020) | 27.29 | 40.14 | 100.00 | 9.90 | 37.90 | 0.00 |

## C  CAN LOGITS DIVERSIFIED INITIALIZATION HELP CIRCUMVENT IMBALANCED GRADIENTS?

Figure 5 shows the GIR values of 5 randomly selected CIFAR-10 test images at the first 20 steps of ODI, FAB, or our MDMT attack. The FAB attack is the most effective attack in the AA ensemble.

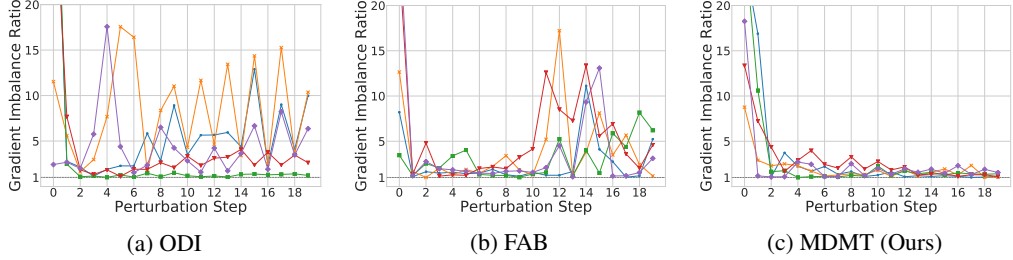

(a) ODI        (b) FAB        (c) MDMT (Ours)

Figure 5: Gradient imbalance ratio at the first 20 steps of ODI (a), FAB (b) and our MDMT (c) attacks on the AdvInterp model for 5 randomly selected CIFAR-10 test images.

## D  CAN RANDOM RESTART OR MOMENTUM HELP CIRCUMVENT IMBALANCED GRADIENTS?

As we discussed in Section 3, many times of random starts can potentially increase the probability of finding an adversarial example. Momentum method is another way to help escape overfitting to local gradients (Sutskever et al., 2013). Here, we test whether random restart or momentum can help avoid imbalanced gradients. For random restart, we run 400-step PGD attack with 100 restarts ($PGD^{100 \times 400}$). For momentum, we use momentum iterative FGSM (MI-FGSM) (Dong et al., 2018) with 40 steps, 2 restarts and momentum 1.0. For both attacks, we set $\epsilon = 8/255$ and step size $\alpha = 2/255$. We apply the two attacks on 1000 randomly chosen CIFAR-10 test images, and report the robustness in Table 6 for the four defense models checked in Section B. Compared to traditional PGD with 40 steps, the robustness can indeed be decreased by $PGD^{100 \times 400}$ except Bilateral, an observation consistent with our analysis in Section 3 that more restarts can lower model accuracy. However, the robustness is still highly overestimated compared to that by our MDMT attack. This indicates that imbalanced gradients can exist in wide-spanned input regions, resulting in a low probability for random restart to find successful attacks. To our surprise, MI-FGSM performs even worse than traditional PGD. On three defense models (eg. Adv-Interp, FeaScatter, and Sense), it produces even higher robustness than PGD. This implies that accumulating velocity in the gradient direction can make the overfitting even worse when there are imbalanced gradients. This again confirms that the imbalanced gradients problem should be explicitly addressed to obtain more reliable adversarial robustness.

Table 6: Robustness (%) of four defense models that have significant imbalance gradients against $PGD^{100 \times 400}$ and MI-FGSM attack.

| **Defense** | PGD | MDMT | $PGD^{100 \times 400}$ | MI-FGSM |
|---|---|---|---|---|
| Adv-Interp | 72.48 | **37.59** | 70.70 | 73.25 |
| FeaScatter | 68.64 | **36.86** | 64.10 | 70.79 |
| Bilateral | 60.95 | **37.21** | 64.08 | 51.52 |
| Sense | 59.86 | **35.25** | 56.00 | 62.41 |

## E  12 EXAMINED DEFENSE MODELS

We focus on adversarial training models, which are arguably the most effective defense models to date. The 12 selected defense models are as follows. The standard adversarial training (SAT) (Madry et al., 2018) trains models on adversarial examples generated by PGD attack. Dynamic adversarial training (Dynamic) (Wang et al., 2019) trains on adversarial examples with gradually increased convergence quality. Max-Margin Adversarial training (MMA) (Ding et al., 2018) trains on adversarial examples with gradually increased margin (*e.g.* the perturbation bound $\epsilon$). For MMA, we evaluate the released "MMA-32" model. Jacobian Adversarially Regularized Networks (JARN) adversarially regularize the Jacobian matrices, and can be combined with 1-step adversarial training (JARN-AT1) to gain additional robustness (Chan et al., 2020). For JARN, we only evaluate the JARN-AT1 as JARN has already been completely broken in (Croce and Hein, 2020). We implement JARN-AT1 on the basis of their released implementation of JARN. Sensible adversarial training (Sense) (Kim and Wang, 2020) trains on loss-sensible adversarial examples (perturbation stops when loss exceeds certain threshold). Bilateral Adversarial Training (Bilateral) (Wang and Zhang, 2019) trains on PGD adversarial examples with adversarially perturbed labels. For Bilateral, we mainly evaluate its released strongest model "R-MOSA-LA-8". Adversarial Interpolation (Adv-Interp) training (Zhang and Xu, 2020) trains on adversarial examples generated under an adversarial interpolation scheme with adversarial labels. Feature Scattering-based (FeaScatter) adversarial training (Zhang and Wang, 2019) crafts adversarial examples using latent space feature scattering, then trains on these examples with label smoothing. TRADES (Zhang et al., 2019) replaces the CE loss of SAT by the KL divergence for a better trade-off between robustness and natural accuracy. Based on TRADES, RTS (Carmon et al., 2019) and UAT (Alayrac et al., 2019) improve robustness by training with $10\times$ more unlabeled data. Misclassification Aware adveRsarial Training (MART) (Wang et al., 2020) further improves the above three methods with a misclassification aware loss function.

## F    GRADIENT IMBALANCED RATIO OF MORE DEFENSE MODELS

In this Section, we provide a complete analysis on the gradient imbalance ratios (GIRs) of all 12 examined defense models and a naturally trained model. The GIR values of these models are shown in Figure 6. One immediate observation is that the GIR value of a defense model is positively correlated with its robustness drop against our MDMT attack in Table 1. Slightly imbalanced defense models SAT, TRADES and RST demonstrate minimum robustness drop, while the PGD-evaluated robustness of highly imbalanced defense models FeaScatter, Bilateral and AdvInterp can drop drastically against our MD attacks. This verifies that higher gradient imbalance can indeed causes more overestimated robustness by regular PGD attack.

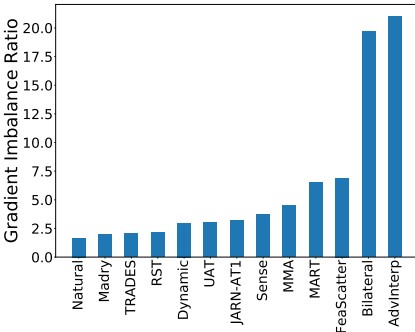

Figure 6: Gradient imbalance ratios (GIRs) of 12 defense models and a naturally trained model ("Natural"). All models are trained on CIFAR-10 dataset.

## G    ABLATION OF THE PROPOSED MD ATTACKS

In this section, we investigate the influence of three factors to our MD attack: 1) initialization method, 2) the second attacking stage, and 3) the stage ordering. We use AdvInterp as our target model, and conduct the following attack experiments on CIFAR-10 test data.

**Initialization Method.** We compare the success rates of our MD attacks using random initialization versus the opposite direction initialization (see Algorithm 1 and Algorithm 2). The results are reported in Table 7. As can be observed, the opposite direction initialization demonstrates a clear advantage over random initialization. Particularly, for MD attack, using opposite direction initialization can improve the attack success rate by 8%, while for MDMT attack, the success rate can also be improved.

**The Second Attacking Stage.** We further investigate the importance of the second stage of attacking with the full margin loss in our MD attacks. Here, we fix the initialization method to the opposite direction initialization. The attack success rates with or without the second stage are also reported in Table 7. We highlight that attacking the full margin loss via the second attacking stage can consistently increase the success rate. Especially for MD attack, a 4.99% improvement can be achieved by the second attacking stage.

**The Ordering of the Stages.** To verify that the ordering of the two stages is suitable for MD attacks, we evaluate a new version of our MD attacks with the two stages are switched: the first stage optimizes the full margin loss and the second stage explores the individual loss terms. The results are reported in Table 7 (the last two columns). As can be observed, MD attacks become much less effective when the two stages are switched. This is because

Table 7: Attack success rates (%) of our MD and MDMT attacks with 1) different initialization methods, 2) with/without the second attacking stage, and 3) with/without stages being switched. Experiments are conducted on defense model AdvInterp and dataset CIFAR-10.

| Attacks | Initialization | | Second Attacking Stage | | Switching Stage | |
| --- | --- | --- | --- | --- | --- | --- |
| | Random | Opposite | without | with | Yes | No |
| MD | 46.32 | **54.67** | 49.68 | **54.67** | 48.41 | **54.67** |
| MDMT | 61.07 | **62.41** | 61.82 | **62.41** | 60.62 | **62.41** |

## H    PARAMETER ANALYSIS OF THE PROPOSED MD ATTACK

We further investigate the sensitivity of our MD attack to two parameters: 1) the number of perturbation steps, and 2) the step size. Here, we focus on the first attacking stage as the second stage is a typical PGD attack, which has been thoroughly investigated in (Wang et al., 2019).

**Number of Steps for the First Stage.** The total number of perturbation steps is set to $K = 40$. When we vary the perturbation steps of the first stage, the remaining steps will be given to the second stage. MD attack will reduce to the regular PGD attack if the perturbation steps of the first stage is set to 0. Here, we vary the steps from 5 to 40 in a granularity of 5. The step size is set to 8/255 and 2/255 for the first and second attacking stage, respectively. The robustness of 4 defense models including Bilateral, Adv-Interp, FeaScatter and Sense are illustrated in Figure 7a. As can be observed, the performance of our MD attack tends to drop at both ends, and the best performance is achieved at $[20, 30]$. Therefore, we suggest to simply use half of the perturbation steps for the first stage (*e.g.* switching to the second stage at the $\frac{K}{2}$-th step).

**Step Size for the First Stage.** We vary the step size used for the first stage from 2/255 to 16/255 in a granularity of 2/255. Following the above experiments, here we fix the number of steps in each stage to 20. The evaluated robustness (or model accuracy on the generated attacks) of defense models Bilateral, Adv-Interp and FeaScatter are illustrated in Figure 7b. A clear improvement of using large step size in the first stage can be observed. Therefore, we suggest to use a large step size for the first stage of exploration.

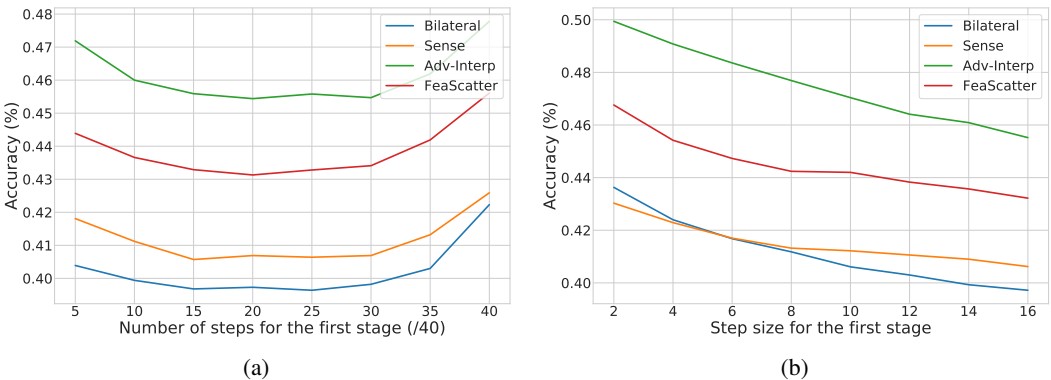

(a)                                         (b)

Figure 7: Parameter analysis of MD attack: (a) the accuracies of 5 defense models under MD attacks with different number of perturbation steps in the first stage; (b) the accuracies of 5 defense models under MD attacks with different step sizes in the first stage.

## I    PARAMETER TUNING FOR PGD ATTACK

In this section, we compare the attack success rates of the PGD attack with different step sizes. We vary the step size from 2/255 to 16/255 in a granularity of 2/255. Note that we only change the step size for the first half of the attacking iterations. The results on 4 defense models including Bilateral, Adv-Interp, FeaScatter and Sense are reported in Table 8. As has also been confirmed in other works (Croce and Hein, 2020; Tashiro et al., 2020), larger step size does help obtain stronger

attacks, especially there are imbalanced gradients (e.g. Adv-Interp, FeaScatter and Sense). However, these finetuned PGD attacks are still far less effective than our MD attacks (see Table 1). On SAT model, the best step size for PGD is 4/255, and larger step size than 4/255 even harms the attack.

Table 8: Adversarial robustness (%) of PGD attack with different step sizes on defense models Adv-Interp, FeaScatter, Sense and SAT trained on CIFAR-10. The results are computed on the entire CIFAR-10 test set. The lowest robustness (i.e. strongest attack) of each defense model is highlighted in **bold**.

| Defense | 2/255 | 4/255 | 6/255 | 8/255 | 10/255 | 12/255 | 14/255 | 16/255 |
|---|---|---|---|---|---|---|---|---|
| Adv-Interp | 72.48 | 72.20 | 71.87 | 71.24 | 70.52 | 69.34 | 67.41 | **65.20** |
| FeaScatter | 68.64 | 67.59 | 66.75 | 66.59 | 64.69 | 63.66 | 61.96 | **59.78** |
| Sense | 59.86 | 58.86 | 58.11 | 57.13 | 56.61 | 55.44 | 54.47 | **53.22** |
| SAT | 45.94 | 45.90 | **46.02** | 46.15 | 46.40 | 46.75 | 47.25 | 47.74 |

## J    COMPARISON TO THE FOUR INDIVIDUAL ATTACKS IN AUTOATTACK

In this section, we compare the model robustness evaluated by the individual attacks in the AA ensemble with our MD attacks. These experiments follow the same setting as in Section 4. The results are shown in Table 9. As can be observed, our MDMT attack demonstrates a superior performance across all the defense models. Moreover, our MD attack which is as efficient as PGD attack can even achieve better performance than all individual attacks on 6 out of 12 models.

Table 9: Attack success rates (%) of the 4 individual attacks (column 2-6) in AA attack and our MD attacks (column 6-7). The best results are highlighted in **bold**. The second best results are highlighted in underline

| Defense | $APGD_{CE}$ | $APGD_{DLR}$ | FAB | Square | MD | MDMT |
|---|---|---|---|---|---|---|
| RST | 61.47 | 60.64 | 60.62 | 66.63 | 60.17 | **59.86** |
| UAT | 59.86 | 62.03 | 58.20 | 66.37 | 59.36 | **56.65** |
| TRADES | 55.08 | 54.04 | 53.82 | 59.48 | 53.10 | **52.78** |
| MART | 55.52 | 52.51 | 51.55 | 57.45 | 51.84 | **51.07** |
| SAT | 46.40 | 46.56 | 46.38 | 53.13 | 45.64 | **45.25** |
| Dynamic | 45.81 | 45.86 | 43.64 | 53.49 | 43.93 | **42.69** |
| MMA | 49.40 | 50.18 | 47.38 | 55.48 | 45.63 | **41.92** |
| Bilateral | 58.26 | 43.11 | 41.36 | 59.07 | 39.82 | **37.21** |
| Adv-Interp | 69.36 | 49.43 | 40.60 | 66.87 | 45.33 | **37.59** |
| FeaScatter | 62.03 | 48.96 | 40.84 | 59.12 | 43.12 | **36.86** |
| Sense | 54.80 | 48.41 | 38.88 | 61.31 | 40.64 | **35.25** |
| JARN-AT1 | 37.25 | 67.55 | 67.40 | 75.32 | 15.03 | **14.60** |

Table 10: Robustness (%) of 3 defense models trained on CIFAR100 data against PGD, AA and our MDMT attacks. The best results (lowest evaluation robustness) are highlighted in **bold**.

| Defense | PGD | AA | MDMT |
|---|---|---|---|
| AT-AWP (Preact ResNet-18) | 30.70 | **25.35** | 25.37 |
| AT-PT (WideResNet-34-10) | 33.5 | **28.42** | 28.45 |
| AT-ES (Preact ResNet-18) | 28.1 | **19.07** | 19.12 |

## K    EVALUATION RESULTS ON CIFAR-100 DATASET

Here, we run our MD attacks on 3 defense models trained on CIFAR-100 dataset: 1) Adversarial Training with Adversarial Weight Perturbation (AT-AWP) (Wu et al., 2020), 2) Adversarial Training with Pre-Training (AT-PT) (Hendrycks et al., 2019), and 3) Adversarial Training with Early Stopping

(AT-ES) (Rice et al., 2020). For AT-AWP and AT-ES, we use their Preact ResNet-18 model, while for AT-PT, we use their WideResNet-34-10 model. The results are reported in Table 10. Our MDMT attack achieved a similar robustness evaluation to the AA ensemble attack, lowering the PGD evaluated robustness by at least 5%. While being as effective as AA attack, our MDMT attacks are more than 8 times faster than AA attack, as we have shown in Table 2.

