# OpenReview forum: "Imbalanced Gradients: A New Cause of Overestimated Adversarial Robustness"
_ICLR.cc/2021/Conference — Reject_

### Official Review · AnonReviewer3 · 2020-10-22
**Didn't quite understand the method, but it looks promising**

**Rating:** 5
**Confidence:** 2

**Review:**

This paper explores constructing adversarial examples in classification, in order to create better robustness metrics for general classifiers. An attack is defined as an epsilon-perturbation of the learned parameters which create a model whose performance is much degraded. The premise of this paper is to use gradient imbalance as a way of creating perturbation targets, which are claimed (and shown numerically) to better fool networks that are trained to withstand more traditional attacks, and can be used to create more robust models in general.

I think the premise is well-motivated, and the results look promising. However, I am having quite a bit of trouble understanding the main method. From the margin loss given, $l(x,y) = z_{\max}-z$, which seems similar to the loss in logistic regression, with gradient  $\nabla \ell(x,y) = yx^T(1-\sigma(yx^T\theta)$. (It is not entirely clear what $z_{\max}$ and $z$ correspond to in the general case.) I am not clear as to why we would expect label balance, e.g. why would $1 = \sigma(yx^T\theta)$? It is also not clear why, when one term dominates the other, that it is a form of nonrobustness.

Alternatively, the interpretation may be that a particular training sample is affecting the model more than the others. That would seem like a more reasonable metric of fallibility, but I don't think that's what the authors intend.

Overall, I do think the paper can be improved if written more clearly, with terms and concepts defined. I guessed that $\Pi$ is projection, and other terms like imbalance ratio are buried in text and hard to find. But, I find the method overall simple and elegant, so with continued discussions with the authors to clarify what exactly is happening, I would raise my score.

---

> ### Author Response · Authors · 2020-11-24
> **Response to AnonReviewer3**
>
> Thanks for your thoughtful comments. Please find our responses below.
>
> ---
> **Q1:** What Zmax and Zy?
>
> **A1:** For multi-class classification, $z$ are the input to the softmax function:
> $p_ {i}(x)=Softmax(z)_ {i}=e^{z_ {i}}/\sum_{j=1}^C e^{z_ {j}}$,
> where $p_ {i}$ is the probability of $i$-th class.
> ---
> **Q2:** meaning of margin loss.
>
> **A2:** Margin loss is a commonly used loss to achieve better attack performance than the conventional cross entropy which may suffer from the vanishing gradient problem during the attack. Margin loss is a good representation of the misclassification adversarial objective [1]. Models are misclassified if and only if margin loss is less than 0.
>
> ---
> **Q3:** Why would $1 = \sigma(yx^T\theta)$?
>
> **A3:** This view is interesting. For adversarial attack, the gradient is computed with respect to the input x, rather than the parameter. The goal of the attack is to slightly perturb x so that prediction is flipped to a wrong one. In addition, imbalanced gradient only occurs when there are multiple output neurons (e.g. at least two logits outputs). Suppose we have two output neurons and two gradient terms (like the one the reviewer suggested, but taken over input x) with each gradient term corresponding to one output neuron. For a given input x, the goal of the attack is to use the gradient information to flip the neuron that currently has the larger value to be the smaller one. If the values of the two gradient terms are not balanced, say one of them is a magnitude higher than the other one, our finding is that attacking the higher magnitude neuron is more effective than attacking the smaller one or both at the beginning.
>
>
> ---
> **Q4:** Why, when one term dominates the other, that it is a form of nonrobustness.
>
> **A4:** When one term dominates the other, the adversarial objective will degenerate to the single dominating loss term, which is no longer a valid objective for misclassification. We have shown in Figure 2(b) that attacking a single loss term will result in a much weaker attack, and thus less reliable robustness evaluation.
>
> ---
> **Q5:** clear writing
>
> **A5:** Thanks for the suggestion. We will add more explanations of the two loss terms and related concepts, and also change the typesetting to make sure that the definition of gradient imbalance ratio is easier to find.
>
> [1] Carlini, N., & Wagner, D. (2017, May). Towards evaluating the robustness of neural networks. S&P.

---

### Official Review · AnonReviewer1 · 2020-10-27
**Interesting hypothesis to be validated, more thorough comparison needed**

**Rating:** 4
**Confidence:** 5

**Review:**

The paper introduces and analyses the possibility that the effectiveness of PGD-based adversarial attacks might be reduced by imbalanced gradients between the terms of the margin losses commonly used. As a remedy, it also proposed a new scheme for PGD attack, where for the first half of the iterations a single-term loss is optimized, before falling back on the the usual margin loss. The authors test the hypothesis of imbalanced gradients, introducing a new metric, GIR, and the newly proposed attack in two versions, MD and MDMT, on several defenses based on adversarial training.

Pros
1. Understanding why some adversarial defenses make some versions of PGD (e.g. with cross-entropy or margin loss) overestimate, sometimes even largely, robustness is an interesting direction, and the proposed explanation of imbalanced gradients is, as far as I know, novel.
2. The proposed attacks are effective against many defenses.

Cons
1. It is not clear what the proposed metric Gradient Imbalance Ratio (GIR) captures and how it is connected the scheme of MD. In fact, the GIR values for JARN-AT1, Sense and MMA, which are not robust, especially against MD, are smaller than for MART and similar to those for UAT, which are instead robust and little or not affected by gradient imbalance according to the results of the MD attacks. Moreover, FeaScatter and MART have similar GIR, but the improvement of MDMT compared to standard PGD is >31% for the former but only around 5% for the latter.
2. In the comparison with other methods (Table 1) the results of AA seem worse than reported in the latest version of (Croce & Hein, 2020), which report lower values of robust accuracy than MDMT for the same defenses. Also, when considering only individual attacks, it'd make sense to include FAB, as in Sec. 4.3 and Appendix C the authors state that it is the most effective in AA and partially avoids the problem of imbalanced gradients.
3. In the MD attack, the first half of the iterations have step size set to $2\cdot \epsilon$, which is than decreased in the second stage. This seems similar to what Auto-PGD does.

Overall, the hypothesis the imbalanced gradients might be a cause of overestimation of robustness is reasonable and worth exploration, but the metric proposed by the authors does not clearly capture this, and the attacks which should overcome that issue are not clearly better than existing methods.

---
Update after rebuttal

I thank the authors for the response. After reading it, the revised version and the other reviews, the concerns expressed in the initial review are still valid.

Then, I keep the initial score.

---

> ### Author Response · Authors · 2020-11-24
> **Response to AnonReviewer1**
>
> Thanks for your valuable comments. Please find our responses below.
>
> ---
> **Q1:** The GIR values for JARN-AT1, Sense and MMA, which are not robust, especially against MD, are smaller than for MART and similar to those for UAT, which are instead robust and little or not affected by gradient imbalance according to the results of the MD attacks.
>
> **A1:** Yes, some of the robust models indeed have larger GIRs than some of the “non-robust” models. GIR measures the imbalance ratio of the gradient, however, it is not a measure for robustness nor should be used as an exact metric to determine whether one model is more robust than the other. Adversarial robustness is a result of many different factors such as model architecture, model capacity, training strategy, training loss, input dimensionality and sample complexity. The models with higher GIR are generally more prone to inaccurate robustness evaluation and are potentially more vulnerable to our MD attacks. For example,  MART is an improved version of TRADES, however, it has a higher GIR than TRADES, which indicates the robustness improvement may be a result of imbalanced gradients. As evaluated in Table 1, it is even worse than TRADES, though only slightly. We have added the discussion to Section 4.2.
>
>
> ---
> **Q2:** AA results
>
> **A2:** The result the reviewer mentioned is the AA+, which is an informal release of AA that includes two more attacks (i.e. targeted APGD-DLR and FAB attacks) into the AA ensemble. The AA attack compared in our paper is the formal version accepted to ICML (Croce & Hein, 2020). Compared to AA attack, our MDMT attack alone can achieve comparable or even better performance with much less computational cost (see our response **A2** for AnonReviewer2).
> Moreover, AA is an ensemble of 4 attacking methods. Comparing our method against such an ensemble method itself is arguably not a fair comparison, even though we still conducted the comparison to the AA ensemble in our paper. Both of our MD attacks are more powerful than any of the attacks in the AA ensemble. The results can be found in the table below.
> Note that our attack can also be combined with the AA attack or even AA+ to form a more powerful ensemble. We have added this comparison and the analysis to Appendix J.
>
> ||APGD_CE|APGD_DLR|FAB|Square|MD|MDMT|
> | ----|:----:|:----:|:----:|:----:|:----:|:----:|
> |RST|61.47|60.64|60.62|66.63|60.17|**59.86**
> |UAT|59.86|62.03|58.20|66.37|59.36|**56.65**
> |TRADES|55.08|54.04|53.82|59.48|53.10|**52.78**
> |MART|55.52|52.51|51.55|57.45|51.84|**51.07**
> |SAT|46.40|46.56|46.38 |53.13 |45.64|**45.25**
> |Dynamic|45.81 |45.86 | 43.64 |53.49 |43.93|**42.69**
> |MMA|49.40|50.18|47.38|55.48|45.63|**41.92**
> |Bilateral| 58.26|43.11 | 41.36 | 59.07 |39.82|**37.21**
> |Adv-Interp|69.36|49.43|40.60|66.87|45.33|**37.59**
> |FeaScatter|62.03|48.96|40.84|59.12|43.12|**36.86**
> |Sense|54.80|48.41|38.88|61.31|40.64|**35.25**
> |JARN-AT1|37.25 |67.55 | 67.40 | 75.32 |15.03|**14.60**
>
>
> ---
> **Q4:** Step size of MD attack.
>
> **A4:** Our MDMT attacks use a fixed step size in both two stages while Auto-PGD uses a complex scheduling strategy to adjust the step size. The reason we choose a large step size in the first stage is according to our comprehensive analysis in Appendix H. On the other hand, large step size is a common technique that has been used in several other attacks [1].
>
> ---
> [1] Tashiro, Y., Song, Y., & Ermon, S. Diversity can be Transferred: Output Diversification for White-and Black-box Attacks. NeurIPS, 2020.

---

### Official Review · AnonReviewer2 · 2020-10-29
**Review report**

**Rating:** 6
**Confidence:** 4

**Review:**

This paper identified the issue of Imbalanced Gradient, verified through some recent defense methods. Motivated by such an issue, a marginal decomposition (MD) attack is proposed to offer a stronger robustness measure. In general, the paper is well written, and the studied problem is interesting. The MD perspective explains why label smoothing may provide insufficient robustness.

My comments are listed below.

1)  In Sec. 3.2, why are the first K/2 iterations used to maximize the individual margin term and then update the entire loss? What will happen if the scheduling is the opposite: updating the entire loss at the first K/2 iterations, then individual terms？Some ablation studies or explanation should be provided.

2) Does the proposed stronger attack offer a stronger min-max defense? Suppose that the ordinary PGD attack is replaced by an MD attack during min-max training, will it offer better overall robustness? The general question to ask is: In addition to root cause analysis on the ineffectiveness of some existing defense methods, what are the additional benefits of the newly proposed MD attack?

3) Does it seem that the MD attack has to run over more iterations than the PGD attack, leading to extensive computation cost?

4) What is the possible effect of the MD attack on the generated perturbation pattern? In the black-box setting, will the MD attack be more query-efficient than a commonly-used PGD black-box attack?


Post-rebuttal:

I am mostly satisfied with the authors' response. After reading other reviewers' comments, I shared a similar concern on the marginal contribution. However, the newly added black-box result is a good addition to the paper. Thus, I keep my original rating toward the positive side.

---

> ### Author Response · Authors · 2020-11-24
> **Response to AnonReviewer2**
>
> Thanks for your thoughtful comments. Please find below our responses to your questions.
>
> ---
> **Q1:** Why are the first K/2 iterations used to maximize the individual margin term and then update the entire loss?
>
> **A1:** As we explained in Section 3.2, the first stage exploits individual loss terms to rebalance the imbalanced gradients, while the second stage ensures that the final objective (e.g. maximizing the classification error) is achieved. Therefore, the entire loss instead of the individual terms should be used at the ending stage, otherwise, the misclassification objective cannot be guaranteed. We have added a new experiment for this ablation study to Table 7 (Appendix G), along with the existing two ablations on different initialization methods and our attacks without the second attacking stage. Other analyses of our method such as number of steps used for the first stage, and the step size for the first stage are provided in Appendix H.
>
> ---
> **Q2:** Can our method help produce stronger min-max defense?
>
> **A2:** Our attacks are proposed to circumvent the imbalanced gradients. Therefore, for defense methods that do not cause imbalanced gradients, training with PGD and our attacks should be the same. Meanwhile, we would like to highlight that the main contribution and focus of this paper is to identify pitfalls in robustness evaluation and promote more reliable robustness evaluation methods (as opposed to developing new defense methods).
>
> ---
> **Q3:** Does it seem that the MD attack has to run over more iterations than the PGD attack, leading to extensive computation cost?
>
> **A3:** No. In our experiments, our MD attack utilizes the same number of iterations as PGD. That is, we divide the iterations of PGD into two stages and apply our MD strategy in the first stage. To have the same time complexity and fair comparisons is the main reason why we did not intentionally increase the perturbation steps of our attacks, though it will further improve our performance. Moreover, we would like to note that our MD attacks are much simpler and significantly more efficient than the current state-of-the-art AA attack. Please find below an efficiency comparison between our attack and AA (AA+ is an informal updated version of AA with two more attacks added into the AA ensemble). We run the attacks for 5 times on the entire CIFAR10 test set and report the average time. We have added this new experiment and the analysis to Section 4.1.
>
>
> ||MDMT|AA|AA+|
> | ----|:----:|:----:|:----:|
> |Adv-Interp|**2.01hrs**|16.36hrs|20.71hrs
> |FeaScatter|**1.97hrs**|15.97hrs|20.44hrs
> |Sense|**2.28hrs**|18.35hrs|23.44hrs
>
>
> ---
> **Q4:** Possible more efficient black-box attack?
>
> **A4:** For gradient estimation based black-box attacks like SPSA, using our MD attacks can indeed help the attack to escape imbalanced gradients, leading to more queries-efficient and successful attacks. We have run additional experiments to confirm this. We use the two-stage losses of our MD attack for SPSA, and denote this method as SPSA+MD. For both SPSA and SPSA+MD, we use the same batch size of 8192 with 100 iterations, and run on 1000 randomly selected CIFAR-10 test images. The robustness of the two methods on Adv-Interp, FeaScatter and Sense models are reported in the table below. Compared to SPSA, SPSA+MD can lower the robustness by at least 5.9%. This is a remarkable improvement, considering the simplicity of the adaptation. We have added this new result to Section 4.3 .
>
> ||SPSA|SPSA+MD|
> | ----|:----:|:----:|
> |Adv-Interp|75.20| **59.70**
> |FeaScatter|71.80|**54.40**
> |Sense|62.10|**51.20**
>
> ---

---

### Official Review · AnonReviewer4 · 2020-11-05
**Review for Imbalanced Gradients: A New Cause of Overestimated Adversarial Robustness**

**Rating:** 5
**Confidence:** 5

**Review:**

This work highlights the existence of imbalanced gradients as a phenomenon that may hinder optimization of gradient-based adversarial attacks and, thus, give a false sense of robustness. Imbalanced gradients may occur as the attack objective consists of the difference of two terms (typically, the outputs of the network on two different classes). When the gradients of these two terms have opposite directions, the attack optimization may get easily stuck in a suboptimal local optimum, thus decreasing the attack effectiveness.
To overcome this issue, the authors propose a simple variant that can also be applied to existing attacks, based on following the gradient of each term in the objective iteratively, rather than considering their combination.

The paper addresses an important issue in adversarial machine learning, i.e., the security evaluation of a defense. In fact, the performance of many proposed defenses has been overestimated due to the weakness of attacks used in the evaluation. One of the causes of this is the inability of gradient-based attacks to correctly perform the optimization.

The paper provides an exhaustive analysis and formulation of the problem of imbalanced gradients, for which also a heuristic metric is provided. This phenomenon is then evaluated on a set of state-of-the-art defenses and attacks.

The authors propose an attack variant inspired from PGD and MT attacks, using the difference of class logits as the objective (i.e., the so-called margin loss). Then, the optimization algorithm works as follows. For each restart, in the first half of the optimization process, only one of the two terms of the margin loss is alternatively used, while in the second half, the whole margin loss is used. The attack formulation and algorithm are clear.
It is important to remark that rather than proposing a new attack framework, the authors actually only propose an optimization variant of the existing PGD and MT attacks and run these attacks with the proposed changes. As correctly stated by the authors, indeed, their proposal can be viewed as an initialization strategy for adversarial attacks (rather than a completely novel attack).

Experiments are reproducible and consider many state-of-the-art attack algorithms and defenses. However, the comparison with other attacks does not seem totally fair. The same hyperparameters (step size, number of iterations and restarts) are used for PGD, C&W and MD attacks (same also for ODI, MT and MDMT), and default hyperparameters are used for the AutoAttack framework. This is problematic, as each attack should be tuned independently, and the choice of hyperparameters should be justified (a common criterion for hyperparameter tuning may be selected, and a grid-search over the attack hyperparameters should be conducted). As shown in many other papers, the attack hyperparameters play a key role in correctly optimizing the attack objective (even single data points can require different hyperparameter tuning).

Another limitation of this work is that it is restricted solely to WideResNet architectures and CIFAR10.

Furthermore, the computational efficiency of the attack is not discussed. Although it is not a primary aspect, it should at least be mentioned when it is compared to other attacks, since it is potentially slower as it might require more iterations and restarts. To be fair, attacks should be even compared under the same complexity (e.g., number of queries to the target model).

In the last column of Table 1, the performance difference between MDMT and PGD is reported. However, this could lead to an overestimation of the attack performance: for a fair evaluation, the MDMT attack should be compared to the stronger attack available at the state of the art, according to the experimental evaluation conducted by the authors, and not to PGD (which has been shown to perform worse than more recent attacks in many papers – e.g., AutoAttack).

---

> ### Author Response · Authors · 2020-11-24
> **Response to AnonReviewer4**
>
> Thanks for your thoughtful comments. Please find below our responses to your questions.
>
> ---
> **Q1.1:** Same hyperparameters for PGD, C&W and MD attacks are not fair.
>
> **A1.1:** We use the same number of iterations and restarts for PGD, C&W and MD attacks so that they can be fairly compared under the same complexity. Also, more restarts and iterations do not necessarily lead to stronger attacks. In Appendix D, we show that 400 steps of PGD attack with even 100 random restarts (overall 40k perturbations for generating one adversarial example) cannot circumvent imbalanced gradients.
> To address your concern, we have conducted the grid search experiment for PGD attack on SAT (standard adversarial training), Adv-Interp, FeaScatter and Sense models. We vary the step size from 2/255 to 16/255 in a granularity of 2/255.The test accuracy (lower test accuracy indicates better attack) results are shown below. While larger step size can be helpful when attacking models with imbalanced gradients (e.g. Adv-Interp, FeaScatter and Sense), the improvement is very limited and our MD attacks are far better than these finetuned PGD attacks. Note that on model SAT, the best step size for PGD is 4/255, and larger step sizes than 4/255 even harm the attack. The results and analysis has been added to Appendix I.
>
> ||2/255|4/255|6/255|8/255|10/255|12/255|14/255|16/255|
> | ----|:----:|:---:|:---:|:---:|:---:|:---:|:---:|:---:|
> |Adv-Interp|72.48|72.20|71.87|71.24|70.52|69.34|67.41|**65.20**
> |FeaScatter|68.64|67.59|66.75|66.59|64.69|63.66|61.96|**59.78**
> |Sense|59.86|58.86|58.11|57.13|56.61|55.44|54.47|**53.22**
> |SAT|45.94|**45.90**|46.02|46.15|46.40|46.75|47.25|47.74
>
> ---
> **Q1.2:** Hyperparameters tuning for AA attack.
>
> **A1.2:** AA attack is claimed to be a parameter-free attack [1], so we think it is reasonable to use its default hyperparameters. Note that the parameters are dynamically adjusted in AA.
>
> ---
> **Q2:** Only WideResNet and CIFAR10.
>
> **A2:** Like existing attacks PGD and MT, our attack is not restricted to particular datasets or models. Please ﬁnd our new results on two CIFAR-100 defense models below. These new results have been added to Appendix K. While being as effective as AA attack, our MDMT attacks are more than 8 times faster than AA attack, as we will show in our next response.
>
> ||Reported|AA|MDMT|
> | ----|:----:|:----:|:----:|
> |AT-AWP (Preact ResNet-18) | 30.70  | **25.35** | 25.37
> |AT-PT (WideResNet-34-10) |33.5| **28.42** |28.45
> |AT-ES (Preact ResNet-18) |28.1|**19.07**|19.12

---

> > ### Author Response · Authors · 2020-11-24
> > **Response to AnonReviewer4**
> >
> >
> > ---
> > **Q3:** Computational efficiency.
> >
> > **A3:** Our two-stage attack can potentially need some additional computation. However, our experiments show that under the same complexity (number of iterations and number of restarts), our MD or MDMT attacks can always achieve a better performance. We have conducted an efficiency comparison between our MDMT attack and AA attack on Adv-Interp, FeaScatter and Sense models. We run the attacks for 5 times on the entire CIFAR10 test set and report the average time cost (in hours) in the table below. This experiment was run on a single 2080TI GPU with the same batch size.
> >
> > Our attack provides a simple but effective solution to avoid the tendency of existing attacks to get stuck in local optima when there are imbalanced gradients, and more importantly, without increasing their total perturbation steps.
> >
> > PGD, CW, and MD attacks (ODI, MT, MDMT) use the same and appropriate number of iterations and restarts. Therefore, they are compared under the same complexity and the comparison is fair. Compared to AA which is an ensemble of four attacks, our MDMT attack alone can achieve the same or even better performance on some defense models with much less computational cost.
> > We have added the efficiency analysis to Section 4.1.
> >
> > ||MDMT|AA|AA+|
> > | ----|:----:|:----:|:----:|
> > |Adv-Interp|**2.01hrs**|16.36hrs|20.71hrs
> > |FeaScatter|**1.97hrs**|15.97hrs|20.44hrs
> > |Sense|**2.28hrs**|18.35hrs|23.44hrs
> >
> > ---
> > **Q4:** Why report the performance difference between MDMT and PGD?
> >
> > **A4:** PGD is arguably the most widely used attack in the literature to benchmark the robustness. By comparing PGD and our MDMT attack, the robustness difference indicates the amount of “overclaimed robustness”. While recent attacks AA and ODI are now being used to test new defense methods, the reason why they are so effective has not been fully understood. Our work provides a gradient perspective understanding of these methods (in section 4.3). This can help motivate even more effective attacks. We have conducted a comparison to the individual attacks used in the AA ensemble on CIFAR-10 dataset. The results are reported in the table below. On almost all tested defense methods, our MDMT is always the best attack. Note that our MD attacks can also be combined into the AA ensemble to further improve its performance. We have added the following results and the analysis to Appendix J.
> >
> > ||APGD_CE|APGD_DLR|FAB|Square|MD|MDMT|
> > | ----|:----:|:----:|:----:|:----:|:----:|:----:|
> > |RST|61.47|60.64|60.62|66.63|60.17|**59.86**
> > |UAT|59.86|62.03|58.20|66.37|59.36|**56.65**
> > |TRADES|55.08|54.04|53.82|59.48|53.10|**52.78**
> > |MART|55.52|52.51|51.55|57.45|51.84|**51.07**
> > |SAT|46.40|46.56|46.38 |53.13 |45.64|**45.25**
> > |Dynamic|45.81 |45.86 | 43.64 |53.49 |43.93|**42.69**
> > |MMA|49.40|50.18|47.38|55.48|45.63|**41.92**
> > |Bilateral| 58.26|43.11 | 41.36 | 59.07 |39.82|**37.21**
> > |Adv-Interp|69.36|49.43|40.60|66.87|45.33|**37.59**
> > |FeaScatter|62.03|48.96|40.84|59.12|43.12|**36.86**
> > |Sense|54.80|48.41|38.88|61.31|40.64|**35.25**
> > |JARN-AT1|37.25 |67.55 | 67.40 | 75.32 |15.03|**14.60**
> >
> >
> > ---
> >
> > [1] Croce, F., & Hein, M. (2020). Reliable evaluation of adversarial robustness with an ensemble of diverse parameter-free attacks. arXiv preprint arXiv:2003.01690.

---

> ### Comment · AnonReviewer4 · 2020-11-25
> **Response the the authors' rebuttal**
>
> I thank the authors for their response and running additional experiments to clarify some points. I appreciate the effort, but I still find the improvement on current attacks too marginal to claim that imbalanced gradients are a significant problem.
>
> When Athalye and Carlini showed the existence of obfuscated gradients, they reported a substantial change in the robustness evaluation of many models. Here we're not in the same case. Improvements are much smaller (around a couple of percentage points) and this leaves me unconvinced that imbalanced gradients are that problematic. Moreover, the metric defined in this work is totally heuristic; thus, there's not even theoretical support behind the existence of imbalanced gradients.
>
> Regarding the stepsize tuning for PGD, why do the authors stop at 16/255? What happens at 18/255? Normally, if the best is achieved at the lower/upper boundary of the parameter values, it's a good idea to test for smaller/larger values too (as we may find better solutions). Moreover, I do not understand if one iteration of each attack corresponds to one query to the target model (i.e., one forward and backward pass). If this is not the case, the comparison should be established on the number of queries (not iterations).
>
> Regarding comparisons on computational complexity: comparing the MD attacks to AA is not fair. AA is a complete framework including 4 attacks. What I meant originally is the overhead that MD attacks add on top of PGD implementations, both in terms of time complexity but also and especially in terms of forward/backward calls on the target model - as the time complexity may depend on the implementation, ability to avoid tensor copies to/from GPUs, etc.
>
> Finally, considering that only L-infinity attacks are considered, I still think that the contribution of this work is too marginal.

---

### Author Response · Authors · 2020-11-25
**Rebuttal Summary**

We want to thank all reviewers very much for the valuable comments and suggestions. We have made the following major updates to address the comments.

---
+ Section 4.1: added a paragraph “Efficiency Analysis” with new efficiency results to compare the time cost of our method to AA (Table 2).
+ Section 4.2: added a paragraph “Correlation between GIR and Robustness” to discuss the relationship between GIR and robustness.
+ Section 4.3: added a paragraph “Black-box Attacks can be Improved by Circumventing Imbalanced Gradients.” along with new experiments to show our MD method can also improve black-box attack SPSA (Table 4).
+ Appendix G: added “The Ordering of the Stages” to show what would happen if the first stage and the second stage of our MD method is switched (Table 7).
+ Appendix I: added “Parameter Tuning for PGD Attack” to show how parameter tuning can help PGD (Table 8).
+ Appendix J: added a new comparison to the four individual attacks in AA ensemble (Table 9).
+ Appendix K: added more evaluation of our method for different networks trained on CIFAR-100 (Table 10).
---
We have revised our paper according to all the valuable comments and please let us know if there is anything still not clear or any other suggestions.

---

### Decision · Program_Chairs · 2021-01-07
**Final Decision**

**Decision:**

Reject

**Comment:**

The paper identifies a subtle gradient problem in adversarial robustness-- imbalanced gradients, which can cause create a false sense of adversarial robustness. The paper provides insights into this problem and proposes a margin decomposition based solution for the PGD attack.

Pros:
- Novel insights into why some adversarial defenses may make some versions of PGD overestimate robustness.
- Proposes a method that is motivated by such findings of imbalanced gradients.
- The proposed attacks are shown effective across a wide range of defenses.

Cons:
- The proposed gradient imbalance ratio could be better motivated: i.e. how is it connected to the scheme of margin decomposition?
- Limited novelty in the attacks: i.e. variant of the existing PGD and MT attacks with some proposed changes.
- Various concerns with experiments (i.e. stepsize tuning, choice of hyperparameters).

Overall, the reviewers felt that there were some interesting ideas and directions presented in the paper; however, the reviewers also felt that the contribution was of marginal significance and more confidence in the various components (i.e. how the proposed metrics measure the imbalanced gradient effect and various concerns in the experiments) would have made the paper more convincing.